# The Changes in Menstrual and Menstrual-Related Symptoms among Japanese Female University Students: A Prospective Cohort Study from Three Months to Nine Months after Admission

**DOI:** 10.3390/healthcare11182557

**Published:** 2023-09-15

**Authors:** Yukie Matsuura, Nam Hoang Tran, Toshiyuki Yasui

**Affiliations:** 1Department of Reproductive and Menopausal Medicine, Graduate School of Biomedical Sciences, Tokushima University, Tokushima 770-8503, Japan; 2Research Center for Higher Education, Tokushima University, Tokushima 770-8502, Japan

**Keywords:** menstrual-related symptoms, female university students, three months and nine months after admission, prospective cohort study, lifestyle

## Abstract

Menstrual and menstrual-related symptoms can significantly impact an individual’s physical and psychological health. Understanding how these symptoms evolve over time is crucial to provide appropriate support and healthcare services to young women. This study aimed to investigate changes in menstrual and menstrual-related symptoms among first-year female university students. A prospective longitudinal design was used to compare the symptom profiles between two time points (three and nine months after admission). Out of 100 female university students, 30 responses were analyzed. Data on menstrual and menstrual-related symptoms were collected using standardized questionnaires focusing on menstrual status and the Menstrual Distress Questionnaire (MDQ); no notable changes occurred between the time points. Approximately half reported having irregular menstruation during the three time periods. Among the sub-scales, premenstrual “impaired concentration” showed a tendency to be lower, whereas menstrual “water retention” tended to be higher in timepoint 2 compared to timepoint 1. “Distractible” was found to be significantly lower in timepoint 2 compared to timepoint 1. There was a significant association between a sleep duration of <7 h and worsened MDQ scores. These findings may underscore the importance of providing comprehensive lifestyle and menstrual education to new university students, along with access to appropriate medical care.

## 1. Introduction

Menstrual and menstrual-related symptoms are commonly experienced by women, impacting their physical, emotional, and social well-being. According to a meta-analysis, the prevalence of menstrual-related symptoms was reported to be 47.8% for premenstrual syndrome (PMS) [1] and that of dysmenorrhea was 71.1% [2]. Menstrual-related symptoms not only disrupt activities of daily living [3] but also have an effect on school life, including academic performance and absenteeism among young women [4,5,6].

The transition to university life represents a significant period of adjustment, and the resulting stress associated with this transition may influence menstrual and menstrual-related symptomatology. Various factors and lifestyle habits have been reported to influence the prevalence and intensity of menstrual-related symptoms, such as diet [7,8,9], sleep [9,10], exercise [8,10,11], smoking and alcohol consumption [12], stress [7,8], internet usage time [8], living alone [13], and dormitory living [7]. When students enter university, they have often left their hometown and live alone. This transition can lead to disrupted lifestyle habits, such as irregular eating and sleeping patterns due to club activities or part-time jobs, increased opportunities for smoking and alcohol consumption, and an overall increase in stress. Therefore, menstrual-related symptoms may increase because of these lifestyle changes.

In a longitudinal study of 85 newly enrolled female university students, Sakuma et al. [14] observed an increase in unhealthy lifestyle habits, such as an increased consumption of fried and snack foods, eating out, and alcohol intake. These habits lead to severe premenstrual symptoms. Participants experiencing sleep deprivation, stress, continued eating out, selective eating, and snack consumption exhibited more severe symptoms. Interestingly, individuals who paid attention to their caloric intake and engaged in light daily exercise experienced stronger symptoms. These findings highlight the impact of unhealthy habits acquired during the university transition period and emphasize the need to provide appropriate health education for incoming female students. However, there seems to be a lack of research specifically investigating changes in the menstrual cycle and menstrual-related symptoms among female university students three months after enrollment.

After examining the lifestyle habits of Japanese freshmen, a notable trend emerged. In Japan, among both male and female university freshmen who commenced their studies in April, a deterioration in breakfast habits was observed between December and February (8–10 months after enrollment) compared to late May [15]. In a cohort study conducted on Australian university freshmen over the course of one year, it was found that the average energy intake significantly decreased at the nine-month mark compared to the baseline [16]. These findings suggest that changes in lifestyle habits, such as dietary habits, may occur around the nine-month mark after entering university, and there may be changes in the severity of menstrual cycles and associated symptoms during this period.

Furthermore, Midzuno [17] conducted a survey on the adaptation of Japanese university freshmen to university life every three months until the end of their second year. The study reported that the lowest adaptation to university life occurred in January of the first year (nine months after admission), which was significantly lower than that in July of the first year (three months after admission). Subsequently, the ninth month after enrollment is a critical time for freshmen, as they are approaching exams and may experience changes in lifestyle habits or insufficient adaptation to university life. Understanding the potential challenges and changes that freshmen face over the nine-month period after enrollment highlights the significance of assessing lifestyle habits, menstrual cycles, and associated symptoms of female university students during such a critical time. Therefore, to gain valuable insights into these aspects, the purpose of this study was to investigate changes in the menstrual cycles and associated symptoms among female university students at three and nine months post-enrollment and to determine their relationship, if any, with lifestyle habits.

## 2. Materials and Methods

### 2.1. Study Design

This study adopted a prospective longitudinal cohort design and used a self-reported web survey for data collection.

### 2.2. Participants

All 100 female students who were admitted to one of the undergraduate courses in April 2022 at a Japanese university located in Tokushima prefecture in Japan were included in the study. Therefore, a sample size calculation was not performed.

### 2.3. Data Collection

Data were collected at two time points: timepoint 1 (15–25 July 2022; three months after admission) and timepoint 2 (16–26 January 2023; nine months after admission). Prior to the remote class, an explanation document was provided to the participants, outlining the research purpose, future methodology, and ethical considerations, including voluntary participation, anonymity, and the freedom to stop answering without grades being affected. Cooperation to participate in the research was also requested. The document included the URL and QR code for the web survey, which was accessed using electronic devices via SurveyMonkey^®^. On the first page of the web survey, participants were asked to confirm their consent to participate by checking a consent checkbox. Although the survey was anonymous, participants were requested to provide their email addresses to receive future survey invitations via email. Email invitation for the timepoint 2 survey was sent only to the participants who responded to the timepoint 1 survey. Each participant was assigned an identification number to track individual changes. This study was approved by the Ethics Committee of Tokushima University Hospital (approval number: 3932-1).

### 2.4. Survey Items

The survey included the following items: Subject attributes: information on the participants’ age, age at menarche, height/weight, history of current gynecological diseases, and use of hormones.Lifestyle habits: The participants were asked to provide information on various lifestyle habits, including breakfast intake frequency, sleeping hours, exercise frequency, smoking, drinking, and living status. We also asked about part-time job employment, participation in circle/club activities, absences due to menstrual-related symptoms, and having a menstrual record.Menstrual conditions: Participants were asked to report their current menstrual conditions, including menstrual cycle length, regularity, duration, and perceived amount of bleeding. They were also asked to provide information about their condition at the time of the first survey and before admission.Menstrual-related symptoms were assessed using the 46-item Menstrual Distress Questionnaire (MDQ). In the current study, we used the Japanese version of the MDQ available with the MDQ License. The MDQ, developed by Moos [18], evaluates menstrual cycle-related physical and mental discomfort. Participants rate the intensity of symptoms during three periods: four days before menstruation, during the menstrual period, and during other periods. The MDQ provides a quantitative comparison of symptom intensity on a scale ranging from 0 (not at all) to 4 (severe). It includes eight subscales that assess the following symptom domains: pain, water retention, autonomic reactions, negative affect, impaired concentration, behavioral change, arousal, and control [19]. In a previous study by Matsuura [20], changes in appetite and food preferences before and during menstruation were observed among Japanese female university students. Therefore, participants were also asked to rate each of three items, including appetite increase and cravings for sweets and snacks, on a scale of 0 (not at all) to 4 (severe).

### 2.5. Statistical Analysis

Each variable was analyzed using descriptive statistics to summarize the data. Cochran’s Q test was used to assess the proportion of women with normal and abnormal menstrual conditions. As most data showed a non-normal distribution, the Wilcoxon signed-rank test was used to analyze changes at two time points for the 46 MDQ items, subcategories, total scores, and three items related to diet. To examine the differences in MDQ total scores between the two surveys, participants were categorized based on the magnitude of the difference, and the Fisher–Freeman–Halton exact probability test was employed to explore the relationship between the number of individuals who changed their lifestyle habits between the first and second surveys. Data analysis was performed using SPSS Statistics version 28.0 for Windows (IBM Corp., Armonk, NY, USA) with a two-sided significance level set at 5%.

## 3. Results

We received 62 responses at timepoint 1 from the 100 female university students invited to participate. At timepoint 2, we received 33 responses from the 62 respondents for timepoint 1, resulting in a total response rate of 33%. Of the 33 respondents, 3 were excluded from the analysis because they had undergone hormone therapy for gynecological diseases prior to university enrollment. Consequently, the analysis was conducted on 30 individuals.

### 3.1. Participants’ Background

Table 1 shows that the mean (±standard deviation (SD)) age of the participants was 18.3 (±0.5) years at timepoint 1 and that the mean age at menarche was 12.0 (±1.5) years. At timepoint 1, the mean height was 158.5 (±5.1) cm, the mean weight was 50.2 (±6.1) kg, and the mean body mass index (BMI) was 19.9 (±1.6) kg/m^2^. Prior to entering university, 16.7% of the participants reported being absent from class during the year due to menstrual symptoms. Between timepoints 1 and 2, out of the six-month absences, one participant (3.3%) experienced premenstrual symptoms and two participants (6.7%) had menstrual symptoms.

### 3.2. Menstrual Status and Changes in Each Timepoint

No significant changes were observed among the students across the three time points in terms of menstrual cycle length, regularity, duration, and perceived amount of bleeding. Approximately half of the students responded that their menstruation was irregular during the three periods (Table 2).

### 3.3. Premenstrual and Menstrual MDQ Subscale Scores at Two Time Points

A comparison of the premenstrual and menstrual MDQ scores at the two timepoints is shown in Table 3. No significant differences were observed in the total MDQ scores or the total scores of the eight subscales between the two timepoints. However, when examining premenstrual symptoms, “impaired concentration” showed a tendency to be lower at timepoint 2 compared to at timepoint 1 (*p* = 0.057). Additionally, among menstrual symptoms, “water retention” tended to be higher at timepoint 2 compared to at timepoint 1 (*p* = 0.090).

Analyzing the individual MDQ46 symptoms, it was found that “distractible” before menstruation was significantly lower at timepoint 2 (median 0.0 (0.0–1.0)) compared to at timepoint 1 (median 0.0 (0.0–1.3)) (*p* = 0.013). During menstruation, “skin blemish or disorder” was significantly higher at timepoint 2 (median 1.5 (1.0–2.3)) compared to at timepoint 1 (median 1.0 (0.0–2.0)) (*p* = 0.041). Furthermore, “distractible” was significantly lower at timepoint 2 (median 0.0 (0.0–1.0)) compared to at timepoint 1 (median 0.5 (0.0–2.0)) (*p* = 0.035).

No statistically significant differences were found in symptoms related to food between the two periods.

### 3.4. Changes in MDQ Score at Two Time Points and Association with Lifestyle Habits

In Table 4, it is shown that the difference in MDQ scores between timepoint 2 and timepoint 1 ranged from −24 to +39 before menstruation and from −26 to +36 during menstruation. These scores were categorized into three groups based on tertials of the MDQ scores: improved, unchanged, and worsened.

Regarding lifestyle habits, the number of individuals who experienced changes in their habits was summarized for the following items: “eating breakfast daily”, “getting 7 or more hours of sleep”, and “engaging in exercise at least once per week”. However, smoking and consuming alcohol were excluded from the analysis, as none of the participants had such habits.

Significant associations were found between changes in the MDQ scores during menstruation at two time points and seven or more hours of sleep (*p* = 0.030). Individuals in the symptom exacerbation group were more likely to not maintain the habit of getting seven or more hours of sleep. However, no significant associations were found at the two time points between lifestyle habits and changes in the MDQ scores before menstruation.

## 4. Discussion

In the current study, the MDQ was employed to gather longitudinal data on menstrual symptoms at two timepoints: three months (timepoint 1) and nine months (timepoint 2) after university admission. Among Japanese university freshmen, no notable changes were observed in menstrual status between timepoints 1 and 2, and there were no significant differences in the total MDQ scores. These findings indicate that even in the ninth month after enrollment, a period when adjustment to university life tends to decline [17], there were no changes in the intensity of menstrual symptoms. However, for premenstrual symptoms, the intensity of “impaired concentration” tended to be lower at timepoint 2 compared to at timepoint 1, and there was a significant decrease in “distractible” both before and during menstruation. In terms of menstrual symptoms, the intensity of “water retention” tended to be higher at timepoint 2 than at timepoint 1. Conversely, during menstruation, “skin blemish or disorder” increased more at timepoint 2 than at timepoint 1. Previous research [14] found that at the three-month mark post-enrollment, menstrual symptoms were more severe than in the immediate post-enrollment period. Specifically, scores in six symptom categories increased significantly, including metabolic, breast, physical, psychological, social, and interpersonal relationship symptoms. However, only psychological symptoms significantly increased during the menstrual phase. Conversely, no significant overall changes were observed between the three-month and nine-month timepoints.

Nevertheless, when comparing timepoint 2 to timepoint 1 after university enrollment, it is conceivable that the stabilization of mental symptoms and enhanced concentration can be attributed to acclimatization to university life. In contrast, there was an inclination for physical symptoms like “skin blemish or disorder” to intensify, but it is plausible to speculate that heightened emotional stability resulted in heightened awareness of bodily changes. Nonetheless, due to the unavailability of initial data upon enrollment for comparison with that of timepoint 1, additional longitudinal investigations are necessary to understand the menstrual symptom changes over time. In a survey conducted in July on Japanese female university students in their first to fourth years using the MDQ, the median MDQ total scores (25th percentile, 75th percentile) were 22.0 (12.0, 41.0) for the premenstrual phase and 34.0 (17.0, 52.5) for the menstrual phase [21]. These findings suggest that significant variations may not occur and that the intensity of symptoms remains relatively similar throughout different academic years and periods.

In this study, the MDQ total scores were categorized into three groups: improved, unchanged, and worsened symptoms. Within the context of changes in these patterns and lifestyle habits during the six-month study period, there was a significant association between the MDQ scores during menstruation and “sleeping for seven hours or more”, with a higher prevalence of insufficient sleep among individuals with worsened symptoms. A meta-analysis indicated that an insufficient sleep duration of less than seven hours is a factor associated with the occurrence of primary dysmenorrhea [22]. A recent systematic review also suggested that women with dysmenorrhea exhibit higher levels of oxidative stress than healthy controls [23]. Endometriosis is a disease primarily associated with chronic pelvic pain caused by the activation of macrophages and mast cells, which contribute to a vicious cycle of persistent inflammation, oxidative stress, and pain [24]. In contrast, melatonin has been reported to have anti-oxidative and analgesic effects [25,26]. It has been suggested that short sleep duration leads to decreased melatonin secretion [27]. Therefore, decreased melatonin secretion due to insufficient sleep may be related to dysmenorrhea. Shortened sleep duration may affect the prevalence of menstrual symptoms. However, the occurrence of primary dysmenorrhea itself reduces sleep duration and dysmenorrhea itself may disrupt normal sleep, although the causal relationship is unclear [10].

No significant association was observed with other lifestyle habits, such as diet and exercise. However, several studies have reported an association between menstrual symptom intensity and lifestyle habits [7,8,9,10,11]. Although it has been reported that first-year Japanese university students tend to practice healthier lifestyles than students in other years [28], our study found that a significant number of students had unhealthy lifestyle habits, such as skipping breakfast, inadequate sleep duration, and lack of exercise. It was also observed that more students engaged in part-time jobs at timepoint 2. As students progress into higher academic years, they may become busier with both academic and personal responsibilities, leading to a deterioration in lifestyle habits that potentially contributes to menstrual abnormalities. Therefore, educational interventions that focus on lifestyle modifications and prioritize healthy habits are needed.

No significant changes were observed in the trends of menstrual cycle length, regularity, duration, or bleeding volume in the pre-admission period, at timepoint 1, or at timepoint 2 post-admission. However, the proportion of students with irregular menstrual cycles was relatively high, accounting for 53.3% before enrollment, 50% at timepoint 1, and 46.7% at timepoint 2. Similar results were reported in a study focusing on female Japanese university students in years 1–4 in which 52.5% of the participants had irregular menstruation [21]. In contrast, a survey conducted between 2001 and 2005 targeting female university students aged 18–20 reported that 33.3% answered that their menstrual cycles were irregular [29]. Therefore, the proportion of participants with irregular menstruation was higher in the present study. Additionally, a study conducted with first-year university students in Turkey found that 60.1% had irregular menstrual cycles [12]. The presence of irregular menstrual cycles is not only indicative of the possibility of underlying organic diseases but also of an increased risk of future cardiovascular diseases, malignancies, and mental disorders [30]. Furthermore, reports suggest that students with irregular menstruation tend to have lower academic performance than those with regular menstruation [31]. In the current study, some might have been unaware of having gynecological conditions such as endometriosis if they had not gone to clinics. Given its potential impact on both future health outcomes and current academic performance, it is essential to provide menstrual health education and encourage students to seek appropriate medical care from healthcare providers. By understanding how sleep duration and other lifestyle factors can impact menstrual health, healthcare providers can develop targeted interventions and educational programs to promote better menstrual well-being among this population. Many promising initiatives on menstrual education and counseling have already been implemented in various countries [32,33,34]. Implementing comprehensive menstrual education during university orientation programs could empower young women with the knowledge and tools to manage their menstrual health effectively. Ultimately, our study’s insights can contribute to improved healthcare services and support systems that enhance the overall well-being and quality of life for young women during their university years and beyond.

This study has several limitations. Owing to the limited sample size of this study, which focused on a single university, generalizing the findings may be challenging. We chose to narrow our investigation to lifestyle-related variables to maintain a clear and targeted research scope. Nevertheless, we recognize that stress and other factors can be important considerations for future research. The amount of menstrual bleeding was determined subjectively; however, several studies employ objective measurements, such as the Pictorial Blood Loss Assessment Chart and Menstrual Pictogram scales, to assess menstrual blood loss and its effects on quality of life [35,36]. We will contemplate incorporating such methodologies in our future investigations. It is important to consider that the research was conducted during a period heavily impacted by the COVID-19 pandemic, with behavioral restrictions and other factors potentially influencing the participants’ lifestyle habits. Moreover, significant seasonal variations in temperature and humidity occurred between July and January. Conducting the second survey in January, which included a winter vacation period and other factors, could have considerably impacted the results. The response rate for the second survey was 53.2%, indicating a significant dropout rate. It is worth considering whether continuing students without problems tend to have higher compliance or whether there is a reverse effect, where students with a stronger interest in health are more likely to participate, potentially introducing bias into the results.

## 5. Conclusions

We found no significant changes in menstrual status or MDQ scores in female university students in Japan at three or nine months after admission. However, these results may enhance our comprehension of the health requirements of students within an academic setting. It has become increasingly evident that fresh-faced students require comprehensive education on lifestyle habits and menstruation, in addition to a support system that ensures prompt access to appropriate medical care when the need arises.

## Figures and Tables

**Table 1 healthcare-11-02557-t001:** Characteristics of participants at each timepoint.

Characteristics	Pre-Admission		After Admission			
			Timepoint 1		Timepoint 2	
			Mean	SD	Mean	SD
Age (years)	NA		18.3	0.5	NA	
Height (cm)	NA		158.5	5.1	158.7	5.1
Weight (kg)	NA		50.2	6.1	50.6	5.4
BMI (kg/m^2^)	NA		19.9	1.6	20.1	1.4
Menarche age (years)	NA		12.0	1.5	NA	
	** *n* **	**%**	** *n* **	**%**	** *n* **	**%**
Absences due to premenstrual symptoms	0	0	NA		1	3.3
Absences due to menstrual symptoms	5	16.7	NA		2	6.7
Have a menstrual record	21	70.0	21	70.0	24	80.0
Have a part-time job	NA		16	53.3	27	90.0
In a circle/club activity	NA		25	83.3	23	76.7

NA: not asked.

**Table 2 healthcare-11-02557-t002:** Longitudinal changes in menstrual status and the occurrence of “normal” state at each survey timepoint.

		Pre-Admission		After Admission				*p*-Value
Menstrual Cycle Status	Category			Timepoint 1		Timepoint 2		
		*n*	%	*n*	%	*n*	%	
Length	25–38 days	27	90.0	26	86.7	27	90.0	0.882
	Other	3	10.0	4	13.3	3	10.0	
Regularity	Regular	14	46.7	15	50.0	16	53.3	0.549
	Irregular	16	53.3	15	50.0	14	46.7	
Duration days	3–7 days	29	96.7	28	93.3	29	96.7	0.717
	Other	1	3.3	2	6.7	1	3.3	
Perceived amount of bleeding	Normal	21	70.0	20	66.7	22	73.3	0.472
	Abnormal	9	30.0	10	33.3	8	26.7	

Cochrane’s Q test.

**Table 3 healthcare-11-02557-t003:** Comparison of MDQ subscale scores between the two timepoints during premenstrual and menstrual phases.

Scales (Number of Items)	Pre-Menstrual Phase					Menstrual Phase				
	Timepoint 1		Timepoint 2		*p*	Timepoint 1		Timepoint 2		*p*
Pain (6)	3.0	(1.0–7.0)	4.0	(2.0–7.3)	0.422	7.0	(4.0–12.3)	8.0	(5.8–12.0)	0.234
Water retention (4)	3.0	(1.8–6.3)	3.5	(2.0–6.0)	0.123	3.0	(1.0–6.0)	3.0	(2.0–6.3)	0.090
Autonomic reaction (4)	1.0	(0.0–2.0)	0.0	(0.0–2.3)	0.534	2.0	(0.0–4.0)	2.0	(0.0–3.3)	0.829
Negative affect (8)	4.0	(1.0–12.8)	4.0	(0.0–11.0)	0.807	5.0	(2.8–11.5)	6.0	(2.8–12.3)	0.677
Impaired concentration (8)	2.0	(0.0–8.3)	0.5	(0.0–6.5)	0.057	4.0	(0.8–10.0)	3.0	(0.0–10.3)	0.156
Behavior change (5)	2.0	(0.0–6.3)	2.5	(1.0–5.3)	0.930	4.0	(2.0–10.3)	3.5	(2.0–9.3)	0.659
Arousal (5)	1.0	(0.0–4.0)	0.0	(0.0–3.0)	0.380	1.0	(0.0–4.3)	0.0	(0.0–3.0)	0.584
Control (6)	0.0	(0.0–1.0)	0.0	(0.0–1.3)	0.242	0.0	(0.0–1.0)	0.0	(0.0–3.0)	0.597
Total MDQ score (46)	21.5	(5.8–47.0)	22.5	(7.0–41.3)	0.991	33.5	(13.8–55.8)	34.5	(15.8–56.3)	0.962

Wilcoxon signed-rank test. The numbers are presented as the median (25th percentile–75th percentile); *p*: *p*-value.

**Table 4 healthcare-11-02557-t004:** Relationship between premenstrual and menstrual MDQ score change patterns and changes in lifestyle habits from the two timepoints.

			Premenstrual MDQ Total Score							Menstrual MDQ Total Score						
			Improved (*n* = 10)		Unchanged (*n* = 10)		Worsened (*n* = 10)			Improved (*n* = 11)		Unchanged (*n* = 10)		Worsened (*n* = 9)		
Total Score Difference			−24~−3		−1~+2		+3~39			−26~−4		−2~+2		+4~36		
Lifestyle Changes between 3 Months and ~9 Months		*n*	*n*	%	*n*	%	*n*	%	*p*	*n*	%	*n*	%	*n*	%	*p*
Eating breakfast daily	Y-Y	13	6	60.0	4	40.0	3	30.0	0.515	6	54.5	3	30.0	4	44.4	0.544
	N-N	11	3	30.0	5	50.0	3	30.0		3	27.3	5	50.0	3	33.3	
	Y-N	3	0	0.0	1	10.0	2	20.0		2	18.2	1	10.0	0	0.0	
	N-Y	3	1	10.0	0	0.0	2	20.0		0	0.0	1	10.0	2	22.2	
Sleep > 7 h/day	Y-Y	6	0	0.0	3	30.0	3	30.0	0.163	2	18.2	4	40.0	0	0.0	0.030
	N-N	14	4	40.0	4	40.0	6	60.0		4	36.4	2	20.0	8	88.9	
	Y-N	2	2	20.0	0	0.0	0	0.0		2	18.2	0	0.0	0	0.0	
	N-Y	8	4	40.0	3	30.0	1	10.0		3	27.3	4	40.0	1	11.1	
Exercising at least once/week	Y-Y	17	6	60.0	7	70.0	4	40.0	0.409	7	63.6	6	60.0	4	44.4	0.646
	N-N	4	1	10.0	2	20.0	1	10.0		1	9.1	2	20.0	1	11.1	
	Y-N	8	2	20.0	1	10.0	5	50.0		3	27.3	1	10.0	4	44.4	
	N-Y	1	1	10.0	0	0.0	0	0.0		0	0.0	1	10.0	0	0.0	

Fisher–Freeman–Halton exact probability test. Y: yes; N: no; *p*: *p*-value.

## Data Availability

The data presented in this study are not publicly available because of privacy restrictions.

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
