# Peer review of "The Changes in Menstrual and Menstrual-Related Symptoms among Japanese Female University Students: A Prospective Cohort Study from Three Months to Nine Months after Admission"

_healthcare, 2023, doi:10.3390/healthcare11182557_

Round 1
Reviewer 1 Report (Previous Reviewer 1)
All of my concerns have been addressed appropriately. However, I would still like to know Why were these two points (3 and 9 months) selected and not any other period
Thank you and well done
Author Response
Please see the attachment.

Reviewer 2 Report (Previous Reviewer 3)
This study is very useful because menstruation has not been adequately studied in Japan. The research is well considered and the content is excellent. I have some concerns below.
The authors have investigated menstrual conditions, but do you know at what age the symptoms start? Some people may also have a disease background such as endometriosis or polycystic ovarian syndrome(PCOS). It would be good to have information on these. If not, please state them in the discussion.
Author Response
Please see the attachment.

Reviewer 3 Report (New Reviewer)
I was pleased to review this paper.
The methodology used by the Authors is appropriate for the purpose of the article description, but but I would like to read materials and methods, for the selection of the works used in the references which are really many.
The English language is fluid and well understood, nevertheless, except in some places that I will select you later, but overall a minor revision is needed.
· For the abstract I found clear the schematization of the content.
· Regarding the introduction you can add these works to deepen the topic DOI: 10.1182/asheducation-2016.1.236; https://doi.org/10.1093/hropen/hoac048.
· The discussion is well described and drawn up.
· Good subdivision into subparagraphs.
· 300-324: You can get better by using more fluid English.
· The structuring of the project is a nice idea but as an expert on the subject, the work would have had more credibility with the use of scores such as the PBAC and MP scales to really define what the cycle of these women is like.
· It would also be nice to do a comparative analysis between the self-perception with respect to the lifestyle of women in comparison with the use of official scales (PBAC/MP).
· I realize that by now the use of what is recommended is not possible, but at least in the discussion I would have access to these stairs as a limitation of the work.
· Explain to me why you preferred to use MDQ total scores.
· Increase the references.
after the few fixes i suggested its a good job.
Author Response
Please see the attachment.

This manuscript is a resubmission of an earlier submission. The following is a list of the peer review reports and author responses from that submission.
Round 1
Reviewer 1 Report
INTRODUCTION
Well written and easy to follow
METHODOLOGY
Please provide a reason for not calculating the sample size
Why were these two points (3 and 9 months) selected and not any other period
Please provide the date of ethical approval obtained
Please provide the name and location of the university as PMS may be influenced by geographical location
What about the different courses/degree taken by the freshmen?
RESULT
TABLE does not include age, BMI, and socioeconomic background. Please include
The 46-item Menstrual Distress 119 Questionnaire (MDQ): Is there a validated version in Japanese? If so, please state.
DISCUSSION
Well written
Author Response
Please see the attachment, thank you!

Reviewer 2 Report
Dear editor:
Thanks for the opportunity to review the manuscript. The chosen topic was highly engaging and intellectually stimulating. However, it is imperative to address the issue of the tables' presentation, as they appear to have been hastily prepared. Moreover, the excessive abundance of tables overwhelms the article, rendering them unsuitable for effective communication of information.
One potential weakness of the article is the small sample size used for analysis. Out of 100 female university students, only 30 responses were included in the study. This limited sample size may not be representative enough to draw generalizable conclusions about first-year female university students as a whole. The findings and associations identified in the study may not accurately reflect the experiences and symptom profiles of a larger population. A larger sample size would have provided more robust and reliable results. The limited size of the sample under investigation poses a significant challenge in terms of generalizability, thereby impeding a comprehensive interpretation and extrapolation of the findings.
Additionally, the manuscript does not provide information on the representativeness of the participants in terms of demographics or other relevant factors. Without this information, it is difficult to determine whether the findings can be generalized beyond the specific group of first-year female university students studied.
It is important to consider other variables that may influence menstrual symptoms, such as stress levels. The abstract lacks a comprehensive analysis of these potential confounders, which limits the depth of understanding regarding the relationship between sleep duration and menstrual symptoms.
While the manuscript briefly mentions the importance of providing comprehensive lifestyle and menstrual education to new university students, it would be valuable to further discuss the potential implications of the findings and how they can inform healthcare services and support for young women.
Quality of English Language is good.
Author Response
Please see the attachment, thank you!

Reviewer 3 Report
The manuscript is a significant paper demonstrating the importance of providing new students with comprehensive lifestyle and menstrual education and the opportunity to receive appropriate medical care.
One point of concern is that these students are female university students, but it seems there is no description of their relationship to the researchers in the text. If they are undergraduate or undergraduate students at the same university as the instructor, is there any bias in the content of the research?
Below are some minor points.
1. The survey Item is easier to understand in the table.
2. How was menstrual volume determined? Is it sensory?
3. Why was "sleep" based on seven hours? Not many people today can sleep for seven hours.
None
Author Response
Please see the attachment, thank you!
